# Long-Term Follow-Up of Patients Needing Extracorporeal Membrane Oxygenation Following a Critical Course of COVID-19

**DOI:** 10.3390/life13041054

**Published:** 2023-04-20

**Authors:** Samuel Genzor, Pavol Pobeha, Martin Šimek, Petr Jakubec, Jan Mizera, Martin Vykopal, Milan Sova, Jakub Vaněk, Jan Praško

**Affiliations:** 1Department of Respiratory Medicine and Tuberculosis, Faculty of Medicine and Dentistry, Palacky University Olomouc and University Hospital Olomouc, 779 00 Olomouc, Czech Republic; 2Center for Digital Health, Faculty of Medicine and Dentistry, Palacky University Olomouc and University Hospital Olomouc, 779 00 Olomouc, Czech Republic; 3Department of Respiratory Medicine and Tuberculosis, Faculty of Medicine, P.J. Safarik University Kosice and University Hospital Kosice, 040 01 Kosice, Slovakia; 4Department of Cardiac Surgery, Faculty of Medicine and Dentistry, Palacky University Olomouc and University Hospital Olomouc, 779 00 Olomouc, Czech Republic; 5Department of Respiratory Medicine and Tuberculosis, Faculty of Medicine and Dentistry, Masaryk University Brno and University Hospital Brno, 601 77 Brno, Czech Republic; 6Department of Psychiatry, Faculty of Medicine and Dentistry, Palacky University Olomouc and University Hospital Olomouc, 779 00 Olomouc, Czech Republic; 7Jessenia Inc. Rehabilitation Hospital Beroun, Akeso Holding, 155 00 Prague, Czech Republic; 8Department of Psychological Sciences, Faculty of Social Sciences and Health Care, Constantine the Philosopher University in Nitra, 949 74 Nitra, Slovakia; 9Department of Psychotherapy, Institute for Postgraduate Training in Health Care, 100 05 Prague, Czech Republic

**Keywords:** ECMO, COVID-19, long-term outcome

## Abstract

Introduction: Severe respiratory failure is one of the most serious complications of coronavirus disease 2019 (COVID-19). In a small proportion of patients, mechanical ventilation fails to provide adequate oxygenation and extracorporeal membrane oxygenation (ECMO) is needed. The surviving individuals need long-term follow-up as it is not clear what their prognosis is. Aim: To provide a complex clinical picture of patients during follow-up exceeding one year after the ECMO therapy due to severe COVID-19. Methods: All subjects involved in the study required ECMO in the acute stage of COVID-19. The survivors were followed-up for over one year at a specialized respiratory medical center. Results: Of the 41 patients indicated for ECMO, 17 patients (64.7% males) survived. The average age of survivors was 47.8 years, and the average BMI was 34.7 kg·m^−2^. The duration of ECMO support was 9.4 days. A mild decrease in vital capacity (VC) and transfer factor (DLCO) was observed on the initial follow-up visit (82.1% and 60%, respectively). VC improved by 6.2% and by an additional 7.5% after 6 months and 1 year, respectively. DLCO improved by 21.1% after 6 months and remained stable after 1 year. Post-intensive care consequences included psychological problems and neurological impairment in 29% of patients; 64.7% of the survivors got vaccinated against SARS-CoV-2 within 12 months of hospitalization and 17.6% experienced reinfection with a mild course. Conclusion: The COVID-19 pandemic has significantly increased the need for ECMO. Patients’ quality of life after ECMO is temporarily significantly reduced but most patients do not experience permanent disability.

## 1. Introduction

Extracorporeal membrane oxygenation (ECMO) temporarily fully substitutes the function of the heart and/or the lung [1]. Blood is drained from the body through a special cannula from a vein (usually femoral or jugular) and returned to an artery (in case of heart failure) or to a vein [2]. The latter configuration is referred to as veno-venous ECMO (vv-ECMO), and it can provide replacement of the oxygenation function of the lungs for the necessary duration. In this way, it is possible to bridge the most critical period in the care of patients with respiratory failure refractory to mechanical ventilation, thus gaining more time for the causal treatment to take effect [3]. The most common indication is extremely severe pneumonia (including COVID-19—coronavirus disease 2019) or bridging in patients waiting for a lung transplant in the terminal phase of lung disease [4]. The majority of such patients suffer from acute respiratory distress syndrome (ARDS). This syndrome is associated with diffuse damage of the alveolo-capillary membrane and frequently leads to development of permanent lung fibrosis [5]. However, the number of studies dealing with long-term follow-up of patients after ARDS is limited [6]. Moreover, the data about the long-term sequelae of the most severe cases requiring ECMO are even more limited [7]. During the COVID-19 pandemic, the use of ECMO has greatly increased due to the large number of critical cases of pneumonia [8]. Thanks to this efficient treatment, a significant proportion of patients with otherwise fatal prognosis survive in the long term, and a new challenge has arisen for pulmonologists—the care of patients after critical pneumonia and ECMO.

## 2. Methods

All patients included in the study required ECMO therapy in the acute stage of COVID-19. The patients were selected for ECMO therapy based on EOLIA criteria [9]. All patients experienced acute respiratory distress syndrome caused by the SARS-CoV-2 virus. The subjects were connected to the ECMO therapy in the University Hospital Olomouc.

After discharge from hospital, all subjects were referred to the Department of Respiratory Diseases of University Hospital Olomouc for thorough examination. The initial visit was performed 3 months after the acute stage of COVID-19 and consisted of detailed anamnesis, physical examination, postero-anterior chest X-ray, and pulmonary function testing, including spirometry, body-plethysmography, and examination of lung diffusion. Pulmonary function tests (PFT) were performed in accordance with the current European respiratory society and American thoracic society guidelines [10]. Body plethysmography MasterScreen by Jaeger^®^ was used for pulmonary function testing, and SentrySuite^TM^ Version 2.19 by CareFusion was used for data retrieval. In individuals with dyspnea and/or reduction in PFT results, a standard 6-min walk test was performed. The predicted 6-min walk distance (6-MWD) was calculated using the formula 800—(5.4 * age). Cardiopulmonary exercise testing using bicycle spiroergometry was performed in unclear cases of dyspnea. High-resolution computed tomography (HRCT) was performed when indicated by the examining physician, and the presence of radiologic changes was evaluated. If the patient required further follow-up, the next check-up was planned three months after the initial examination, followed by a further check-up at one year post-COVID-19. Chest X-ray and PFT were performed during the follow-up visits. Based on clinical necessity, some patients underwent more frequent check-ups and/or the follow-up period was prolonged. Other examinations (neurology, polysomnography, psychology/psychiatry, etc.) were indicated in selected individuals based on clinical necessity. All the patient data, including acceptance of vaccination against COVID-19 and reinfections, had been recorded. If patients were unable or unwilling to attend the follow-up visit in person, a phone call was scheduled to retrieve information about the patient’s condition 3 months after the acute phase, followed by a phone call at six months and one year post-COVID-19.

All subjects agreed to join the study and either gave written informed consent or verbally consented (via telephone) to the use of their data. The local ethical committee approved the study (Ethical Committee of University Hospital Olomouc and Faculty of Medicine and Dentistry of Palacky University Olomouc, Czech Republic) with decision number 98/21 (date 7 June 2021).

## 3. Results

Of the 41 patients indicated for ECMO life support at the Department of Anesthesiology, Resuscitation, and Intensive Care of University Hospital Olomouc, 17 patients (11 males) survived hospitalization, and 16 of these patients were subsequently followed-up at the Department of Pulmonary Diseases and Tuberculosis of University Hospital Olomouc. Each patient from the surviving cohort received systemic corticosteroid treatment in the acute phase. Eight of them (47%) were also treated with remdesivir, one received tocilizumab (with corticosteroids), and one other patient had a treatment course with baricitinib (with corticosteroids and remdesivir). The course of hospitalization was complicated for most patients. Six of them experienced urinary tract infections, and all had developed ventilator-associated pneumonia (in all cases of multi-microbial etiology). Myopathy of the critically ill developed in three of the patients. One patient developed sacral decubitus, and renal failure requiring continuous renal replacement therapy also developed in one individual. Surprisingly, only one patient had contracted *Clostridium* infection; however, it should be noted that all patients received low-dose enteral vancomycin prophylaxis (a single dose of 125 mg daily). One patient died prior to follow-up, and two patients refused to be examined in our center. The latter two patients were contacted by phone and stated that their condition was very good and that they had no limitations in daily life. Only two patients required corticosteroid treatment following hospitalization. One patient received 20 weeks of therapy with systemic glucocorticoids, and one patient received one year of treatment with inhaled glucocorticoids. The rest of the patients underwent respiratory physiotherapy according to the local standards. Ten patients completed the above-mentioned follow-up, which consisted of at least three visits. Seven of them underwent more than three visits (four individuals had four visits, two had five visits, and one had six visits). Two patients were seen twice, and three patients were seen once. All patients participated in phone calls as described above to provide information about their physical condition during the first year after the acute stage of COVID-19. 

Most of the patients (81%) were obese. Only one patient had a normal body mass index (BMI) of 20.8; two individuals were overweight (BMI 25 to <30), six patients had class 1 obesity (BMI 30 to <35), two patients had class 2 obesity (BMI 35 to <40), and three patients had morbid obesity (BMI above 40, with a maximum of 58.6).

The basic characteristics of long-term ECMO survivors are listed in Table 1.

The most common comorbid disease was hypertension (n = 9), followed by diabetes (n = 4). The mean number of comorbid diseases was 1.7; the maximum was 6, and four patients had no comorbid illness. Two females were pregnant at the time of ECMO support; both were in an advanced stage of pregnancy, which necessitated Caesarean section. Both children born by caesarean section were viable at birth and are now (25 months later) healthy. Of all patients monitored, only 31% had a positive history of smoking. Three individuals were active smokers before the acute phase of COVID-19, and all stopped smoking immediately after discharge. Two patients were former smokers at the time of admission, and the rest of the patients were non-smokers. Eight patients had no partner and lived alone or with their parents/family. The rest lived together with their partner and/or children. The list of comorbid diseases and conditions is summarized in Table 2.

The first follow-up visit (via telephone for three patients) was scheduled three months following hospital discharge. Four patients reported no dyspnea (NYHA 1), 10 had mild exertional dyspnea (NYHA 2), and 3 had severe exertional dyspnea (NYHA 3). None had dyspnea at rest. The mean NYHA score was 1.94 at the initial examination. All patients had at least partially improved their exercise tolerance by follow-up. During the check-up after three months, NYHA 1 and 2 was reported in 9 and 7 patients, respectively, with one patient reporting NYHA 3. The mean NYHA score after three months of follow-up was 1.53. Further improvement was observed after one year, when ten patients reported NYHA 1 and 7 reported NYHA 2, with a mean NYHA score of 1.41. The development of NYHA score is displayed in Figure 1.

Pulmonary function tests (PFT) at the initial visit showed a mild decrease in vital capacity (VC) in four individuals and a moderate reduction in vital capacity in one patient, with normal values of VC in the remaining subjects. Forced exhaled volume in 1 s (FEV1) was mildly decreased in three individuals, with all other subjects reaching FEV1 values within the normal range. Total lung capacity (TLC) was mildly reduced in four individuals, and residual volume (RV) was decreased in only one patient. Lung diffusion capacity was reduced in the transfer factor of almost all patients at the 3-month visit except for two with normal values. Two individuals had a severe decrease in transfer factor (DLCO), six had a moderate reduction in DLCO, and three had a mild decrease in DLCO. Transfer coefficient (KCO) was reduced moderately in three individuals and mildly in the other three individuals. Six-minute walk test distance (6MWD) was decreased in 8 patients, mostly mildly, with one patient exhibiting severe reduction in 6MWD (32% of the predicted value). However, the 6MWD of this particular patient improved to 72% of the predicted value after nine months.

Detailed results of the initial PFT are listed in Table 3. 

Most PFT values had improved in almost all patients at subsequent follow-up visits. However, in three patients, a mild decrease in VC (up to 240 mL) was found together with an increase in body weight. Transfer factor and transfer coefficient increased consistently in all individuals at both 6 months and one year after the acute stage of COVID-19. More detailed results are listed in Table 4. Note that the mean value of percentage of predicted diffusion capacity was lower in visit three compared to visit two, likely due to two subjects with very good PFT results refusing to complete the third follow-up visit. The radiological findings improved over time in all subjects. Extensive residual post-inflammatory changes were still present in 4 individuals at the 1-year follow-up, with only unremarkable and bland stripes being apparent in the lung parenchyma on chest X-rays of the remaining patients. HRCT of the chest was performed in 9 subjects 1 year after discharge, revealing mostly mild post-inflammatory changes in the lung parenchyma. Extensive fibrotic changes (reticulations, distortion of lung parenchyma and traction bronchiectasis) were apparent in four patients. Six patients showed signs of air-trapping on HRCT.

Five patients required psychiatric examination. Two of these patients were newly diagnosed with major depression requiring anti-depressive therapy; one patient was diagnosed with schizoaffective disorder, and anti-psychotic medication was prescribed. The psychiatric condition of all three patients improved after the anti-depressive treatment. Another patient remained disabled due to decompensation of known bipolar disorder, and repeated admission to the psychiatry ward was needed. The patient’s mental condition did not significantly improve during the one year of follow-up, making the likelihood of permanent disability probable. The last of mentally affected patients reported an unresolved decrease in short-term memory and shortened sleep duration, but the psychology and psychiatry examination did not show any pathology requiring therapy.

A number of patients (n = 13; 76,4%) complained of neurological symptoms, requiring neurology examination with electromyography (EMG) or electroencephalography. The most common complaint was skin sensitivity changes, especially in the femoral region. Similarly common was the presence of paraesthesia of lower extremities. However, the results of needle EMG were pathological only in a minority of the patients. The results of the examination and typical complaints or pathological conditions are listed in Table 5. 

A sleep study was performed at night on four individuals because of suspected disordered breathing during sleep. One subject had a completely negative polysomnography reading, two had mild sleep apnea syndrome without the indication for positive airway pressure therapy (PAP), and one patient was diagnosed with moderate sleep apnea syndrome (apnea-hypopnea index 24) and started treatment with continuous positive airway pressure therapy. The treatment was well tolerated and led to an improvement in the subject’s sleep. 

None of the subjects were vaccinated against COVID-19 prior to the infection that warranted the use of ECMO, but 11 of the surviving patients (i.e., 64.7%) decided to get vaccinated after overcoming the disease, while six patients (i.e., 35.3%) had received no vaccine against COVID-19 until 1 March 2023. Two patients received a single dose (Johnson vaccine), while two and five patients received two and three doses, respectively. Most individuals received Pfizer vaccines (7), and three received the vaccine by Moderna. Three individuals (i.e., 17.6%) experienced a mild (outpatient management) COVID-19 reinfection. One female patient (aged 37) experienced a severe course of influenza (H1N1) during follow-up, requiring hospitalization with high-flow oxygen support. The condition of this patient was complicated by acute respiratory distress syndrome and sepsis, requiring systemic glucocorticoid administration. After the discharge, the doses of glucocorticoids were gradually tapered and were discontinued two weeks later. One male patient (aged 60) developed tracheal stenosis after prolonged tracheostomy (72 days), which required surgical intervention (resection and end-to-end anastomosis). No other serious event was recorded in any of the individuals at the follow-up. 

At the end of the one-year follow-up, two female patients were on maternity leave, two were retired, and two individuals remained incapable of work and were candidates for a disability pension. The rest of the patients were working again and reported no major limitations in daily life.

## 4. Discussion

Since the beginning of the COVID-19 pandemic, there has been a gradual increase in the number of publications related to post-COVID complications in various groups of patients. Overcoming SARS-CoV-2 infection is often accompanied by subsequent symptoms, primarily shortness of breath, fatigue, pain, and mental problems. Additionally, there is also a temporary decrease in lung functions [11,12,13]. A more objective assessment of these patients using cardiopulmonary exercise testing (CPET) indicates that shortness of breath in the context of post-COVID complications may be mainly due to deconditioning and that the percentage of patients with reduced exercise tolerance increases with the extent of residual changes on the chest X-ray [14,15]. However, only limited data are available on long-term follow-up of patients with critical COVID-19 requiring ECMO support. An observational study by Steinbeis et al. [16] showed that even one year after the acute phase of the disease, significant deficits were present in the functional examination of the lungs in 16 patients. The probability of developing a ventilatory defect after ECMO was even slightly higher than in the case of patients after invasive ventilation support without ECMO (adjusted odds-ratio 7.8 for ECMO vs. 10.5 for mechanical ventilation support); of course, there may be bias in the form of the survivor effect (a large proportion of the critical patients on ECMO do not survive and therefore cannot be further clinically examined). According to the St. George’s Respiratory questionnaire, the quality of life was significantly better in the 12th month of follow-up than in the sixth week. Paradoxically, patients after a mild course of COVID-19 showed practically no changes in their complaints throughout the follow-up period. Ego et al. [17] published the results of 9 patients after ECMO in the critical course of COVID-19. In the study population, survival to discharge was reported in 34% of patients connected to vv-ECMO (11 out of 32). In seven of the nine patients amenable to further investigation, a deficit in lung diffusion capacity for carbon monoxide was demonstrated (median DLCO was 58% of the predicted values). A median distance of the six-minute walking test of 468 m was observed, representing 68% of the predicted values. Nevertheless, none of the patients showed latent respiratory insufficiency (hemoglobin oxygen saturation after exercise was 91 to 96%). This is in concordance with our study, where none of our patients suffered from chronic respiratory insufficiency. Grasselli et al. [18] published a cohort of patients after acute respiratory distress syndrome (ARDS) from any cause, with 34 patients indicated for ECMO and 50 patients indicated for invasive mechanical ventilation (IMV) without ECMO.

Survival at the 12th month of follow-up was 66% in patients after ECMO support and 59% after IMV alone. The differences were not significant. The results of pulmonary function tests (spirometry, body plethysmography and diffusion capacity) were almost within normal limits after 12 months, and no significant differences between the groups were demonstrated. Surprisingly, worse quality of life was consistently reported in the post-IMV group throughout the follow-up period. Long-term follow-up of lung functions shows that the rate of decline in vital capacity as well as DLCO increases with the severity of the disease and the intensity of respiratory support required to manage the acute disease, and patients treated with ECMO have a significant disability. Comparing patients treated with ECMO or IMV and interpreting the differences in the data is often not easy, in part due to the different age distribution of the groups and the selection of patients during the peak of the pandemic. The good news is the fact that despite the DLCO reduction persisting even after one year in some of the patients, most of the patients experience a significant improvement over time, which is also in line with our findings, where the improvement occurred during the first six months of follow-up and remained stable at one year. Patients on invasive ventilation for ARDS are at risk of developing ventilator-induced lung injury (VILI). [19]. Despite ECMO usually being the last resort for patients with the most severe ARDS (or other cause of respiratory failure refractory to IMV), the available data show an overall good long-term prognosis for ECMO survivors. One of the explanations for these favorable outcomes is also the assumption that ECMO makes it possible to bridge the most serious phase of the disease, to stabilize the condition without the need to escalate IMV support by enabling lung protective ventilation [20]. 

Systemic inflammatory response is a major etiopathogenetic mechanism of severe lung damage in individuals with SARS-CoV-2 [21]. Each patient from our cohort received systemic corticoid treatment, eight of them were also treated with antivirotics (remdesivir), and the rest of them were not indicated for specific treatment because of longer delay from initial symptoms. Two individuals also received other immunosuppressant medication (tocilizumab and baricitinib). Similarly, lung regeneration is the reason for radiological and functional restoration after ECMO. However, this process is only partly known, and it is very likely affected by several factors (e.g., differential expression of pathways) that could explain interindividual variability in recovery after damage [22].

It is known that practically all patients with a severe course of COVID-19 have not been vaccinated and that vaccines reduce the risk of a complicated course of the disease [23]. On the other hand, little information is available regarding the acceptance of vaccination against SARS-CoV-2 after overcoming COVID-19. One such study (Seeßle et al., 2021) enrolled 96 patients (32.3% requiring hospitalization during the acute illness) with complaints of post-COVID symptoms. Only 8 (8.3%) of these patients opted for vaccination within 12 months of follow-up [24]. In contrast, another cohort of 97 survivors of critical COVID-19 (Gonzáles et al., 2022) has seen a vaccination rate of 82.3% (79 subjects) within the first year of follow-up [25]. We could therefore say that the acceptance of vaccination is higher in subjects who experienced a more severe course of COVID-19. The data from our cohort confirm these results, with the vaccination rate as high as 64.7% among the survivors of critical COVID-19 requiring ECMO support. This is finally documented by our unique set of patients, where up to 64.7% of patients who survived hospitalization with ECMO support received the vaccine. After overcoming COVID-19, immunity against re-infection increases, which can persist for more than a year. COVID-19 reinfections are relatively rare, reaching 0.66% after more than 12 months according to the meta-analysis by Flacco et al. (2022) [26]. Vaccination further reduces the risk (0.32% among vaccinated subjects vs. 0.74% for unvaccinated individuals). In our cohort, three out of 17 patients (17.6%) experienced COVID-19 reinfection within the first year of follow-up. All reinfections had a mild course and were managed on an outpatient basis, demonstrating a good immune response and the potential protective effect of vaccination.

Five patients required psychiatric care, and it is common for patients to present with variable psychological and psychiatric problems after ECMO. The study by Khan et al. [27] concluded that ECMO survivors have higher rates of psychiatric morbidity and worse quality of life on average. However, these rates are comparable to other severely ill patients. In the study by Risnes et al. [28] on 28 survivors of ECMO, the most common newly diagnosed psychiatric conditions were organic mental disorders, including mood disorders and OCD. This is partly in accordance with our results, as from the five patients in our study, 4 presented different mood disorders (2 with depression, 1 with schizoaffective and 1 with bipolar disorder).

Moreover, Park et al. [29] analyzed data from 3055 ECMO survivors and found that post-ECMO rates of depression were significantly higher compared to the pre-ECMO depression group. Additionally, they discovered that post-ECMO depression was associated with higher all-cause mortality. Risnes et al. also presented more elevated rates of depression evaluated by psychometric scales. In a more recent robust study of 642 ECMO survivors, Fernando et al. [30] proved that ECMO survivors had a higher risk of developing mental disorders than ICU patients. 

The worldwide acceptance of COVID-19 vaccination was estimated to be around 66% [26], which closely coincides with the vaccination rate observed in our small cohort. Interestingly, despite the relatively low vaccination rate and persistent risk factors (obesity, comorbid diseases), none of the subjects experienced a severe reinfection. Luckily, it seems that survivors of severe COVID-19 develop a potent and lasting immune response to SARS-CoV-2, which is especially true for those who get vaccinated following the acute infection [31,32,33,34,35].

The prophylaxis of Clostridium by peroral low-dose vancomycin was performed in accordance with guidelines of the European Clinical Microbiology society [36]. The clostridial infection was proven only in one subject, which illustrates that such treatment should be considered in multimorbid individuals receiving multiple antibiotic treatment.

Strengths and limitations of the study: The main strength of the study is the detailed description of a variety of parameters and complications of critical courses of COVID-19 requiring ECMO support. On the other hand, the study cohort is unicentric and relatively small. Moreover, we did not use standardized questionnaires. In addition, we do not have full data about all of the individuals who did not survive, as many of them were patients referred from peripheral hospitals, and the data is missing. However, we can provide the data we have to other researchers for inclusion in meta-analyses. Moreover, according to the best of our knowledge, other studies that would offer such complex follow-up of ECMO survivors in the context of critical COVID are not available. 

## 5. Conclusions

Extracorporeal membrane oxygenation (ECMO) is a viable treatment opportunity for patients with otherwise refractory respiratory failure. The correct selection of the patient (considering the indication of the treatment and the probability of its effectiveness according to the scoring systems) also significantly increases the therapeutic success in the treatment of ARDS. Patients’ quality of life after ECMO may be significantly reduced, but most do not experience permanent disability. The onset of the COVID-19 pandemic has led to a significant increase in the need for ECMO, which has led to an increase in the use of this treatment modality among anesthesiologists and intensivists. The care of long-term survivors of ECMO poses a new challenge for pulmonologists.

## Figures and Tables

**Figure 1 life-13-01054-f001:**
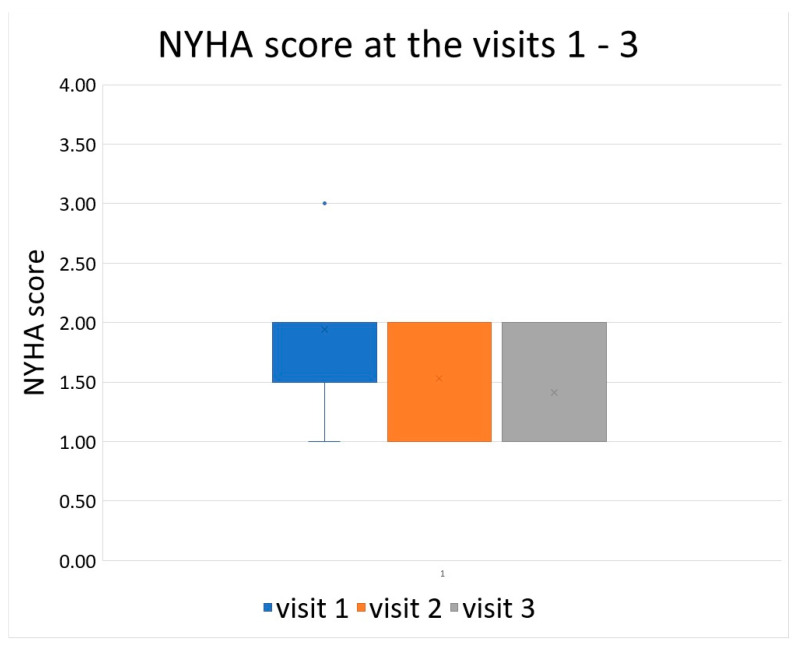
Development of NYHA score between the visits.

**Table 1 life-13-01054-t001:** Basic characteristics of long-term ECMO survivors.

Parameter	Mean	SD
Age (in the acute phase)	47.8	15.0
BMI	34.7	8.0
Hypoxemic index at ECMO connection	76.6	16.3
Quick SOFA score at time of admission	2	0.8
Number of days from positive COVID test result to admission	3.2	3.2
Days spent in hospital before orotracheal intubation	4.5	4.2
Days spent on mechanical ventilation before ECMO	1.9	1.1
Days spent on ECMO support	9.4	2.3
Days spent on mechanical ventilation after ECMO	9.3	14.8
Days spent on high-flow oxygen support after ECMO	2.2	2.9
Days spent on low-flow oxygen support after ECMO	8	8.2

Abbreviations: BMI—body mass index; ECMO—extracorporeal membrane oxygenation; COVID—coronavirus disease.

**Table 2 life-13-01054-t002:** List of comorbid diseases and conditions and their prevalence among ECMO survivors.

Comorbid Disease/Condition	n	%
Hypertension	9	52.9
Diabetes	4	23.5
Hypothyroidism	4	23.5
Hyperlipidemia	4	23.5
Pregnancy	2	11.8
Smoking	3	17.6
Former smoking	2	11.8

**Table 3 life-13-01054-t003:** Pulmonary function test results at the initial examination (n = 15).

Parameter	Mean	SD
VC (% of predicted)	82.1	10.4
FEV_1_ (% of predicted)	87.8	11.2
TLC (% of predicted)	87.6	14.2
RV (% of predicted)	103.0	27.5
DLCO (% of predicted)	60.0	19.9
KCO (% of predicted)	81.3	18.7
6MWT distance	353.8	94.4
6MWT distance (% of predicted)	66.7	20.0

Abbreviations: VC—vital capacity; FEV_1_—forced expiratory volume in 1 s; TLC—total lung capacity; RV—residual volume; DLCO—diffusing capacity of the lungs for carbon monoxide (transfer factor); KCO—transfer coefficient; 6MWT—6-min walk test.

**Table 4 life-13-01054-t004:** Pulmonary function test results in development at check-ups 2 and 3.

Parameter	Visit 2 (6 Months)	Mean Improvement vs. Visit 1	Visit 3 (1 Year)	Mean Improvement vs. Visit 2
Number of Patients Completed Visit	n = 12		n = 10	
VC (% of predicted)	86.8	250 mL (6.2%)	92.1	210 mL (7.5%)
FEV1 (% of predicted)	92.5	170 mL (5.1%)	99.6	300 mL (8.7%)
TLC (% of predicted)	87.2	180 mL (3.2%)	94.8	0 mL (0%)
RV (% of predicted)	97.1	50 mL (1.3%)	105.0	210 mL (8.5%)
DLCO (% of predicted)	70.9	21.1%	70.6	3.8%
KCO (% of predicted)	93.8	18.6%	90.6	3.5%

Abbreviations: VC—vital capacity; FEV_1_—forced expiratory volume in 1 s; TLC—total lung capacity; RV—residual volume; DLCO—diffusing capacity of the lungs for carbon monoxide (transfer factor); KCO—transfer coefficient.

**Table 5 life-13-01054-t005:** Reported neurological complaints and their confirmation.

Complaint/Condition	Number of Reporting Patients	Confirmation of Diagnosis
Epilepsy	1	1
Decreased cutaneous sensitivity	6	2
Paraesthesia	6	2

## Data Availability

The data presented in this study are available on request from the corresponding author. The data are not publicly available due to the privacy of the patients.

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
