# Peer review of "Long-Term Follow-Up of Patients Needing Extracorporeal Membrane Oxygenation Following a Critical Course of COVID-19"

_life, 2023, doi:10.3390/life13041054_

Round 1

Reviewer 1 Report

The authors in their manuscript "Real-life long-term follow-up of patients surviving critical course of COVID-19 with need of extracorporeal membrane oxygenation", described very well the follow-up of ECMO survived COVID-19 patients and the need for the use of this techique in the management of patients refractory to to conventional lung-protective ventilatory support

Author Response

Dear reviewer, 

thank you for your positive review report. We appreciate it very much.

Kind regards.

Reviewer 2 Report

Thank you for the opportunity to review this work. The Authors performed a well-designed cohort study reporting on the outcomes of patients who underwent ECMO due to severe COVID-19. The paper is scientifically sound, even if it is only descriptive, but I think it can be informative to the readers.

I have some comments:

1.  Please make explicit the study design (e.g. cohort) at the beginning of the methods section.

2. Some baseline data are required, especially those about the severity of patients' condition (e.g. PaO2/FiO2 and/or WHO clinical progression scale score and/or radiological scores for parenchymal involvement). Indeed, these can be related with outcomes.

3. Please report data about pharmacological therapies during hospitalization, at least those with a highly proven efficacy in COVID-19 e.g. glucocorticoids. Please also note that different levels and patterns of inflammation, which can be targeted by different doses ad durations of anti-inflammatory treatment can explain different outcomes and different functional characteristics at follow-up. This should be discussed and referenced (Salton F et al. Cytokine Profiles as Potential Prognostic and Therapeutic Markers in SARS-CoV-2-Induced ARDS. J Clin Med. 2022)

4. Similarly, lung regeneration is the reason for radiological and functional restoration after ECMO. However, this process is only partly known and it is very likely affected by several factors (e.g. differential expression of pathways) that could explain interindividual variability in recovery after damage. This should be discussed and referenced (Baratella E et al. Radiological-pathological signatures of patients with COVID-19-related pneumomediastinum: is there a role for the Sonic hedgehog and Wnt5a pathways? ERJ Open Res. 2021)

5. Tables 1 and 3: please report only mean or median (depending on the distribution of the variable) and the respective dispersion index (SD or IQR).

Author Response

Dear reviewer, thank you for your valuable review report. We adjusted the article according to your recommendations, which helped to improve article quality. Following part is aiming to answer or comment on your remarks.

1.Please make explicit the study design (e.g. cohort) at the beginning of the methods section.

The description of the study design was improved

  1. Some baseline data are required, especially those about the severity of patients' condition (e.g. PaO2/FiO2 and/or WHO clinical progression scale score and/or radiological scores for parenchymal involvement). Indeed, these can be related with outcomes.

Baseline data about the patients were added in the results (in table 1), unfortunately, only part of the documentation is available – we have data about quick-SOFA score at the admission and Hypoxemic/Horowitz index before the ECMO-connection

  1. Please report data about pharmacological therapies during hospitalization, at least those with a highly proven efficacy in COVID-19 e.g. glucocorticoids. Please also note that different levels and patterns of inflammation, which can be targeted by different doses ad durations of anti-inflammatory treatment can explain different outcomes and different functional characteristics at follow-up. This should be discussed and referenced (Salton F et al. Cytokine Profiles as Potential Prognostic and Therapeutic Markers in SARS-CoV-2-Induced ARDS. J Clin Med. 2022)

Therapy during the hospitalization is now described in separate paragraph. Discussion was also adjusted according to your recommendation.

  1. Similarly, lung regeneration is the reason for radiological and functional restoration after ECMO. However, this process is only partly known and it is very likely affected by several factors (e.g. differential expression of pathways) that could explain interindividual variability in recovery after damage. This should be discussed and referenced (Baratella E et al. Radiological-pathological signatures of patients with COVID-19-related pneumomediastinum: is there a role for the Sonic hedgehog and Wnt5a pathways? ERJ Open Res. 2021)

Discussion was adjusted according to your recommendation.

  1. Tables 1 and 3: please report only mean or median (depending on the distribution of the variable) and the respective dispersion index (SD or IQR).

Tables were adjusted

Reviewer 3 Report

In their work, Genzor et al. performed a small observational cohort study on 17 survivors of previous veno-venous ECMO treatment due to COVID-19 associated ARDS. The surviving patients were than observed over the period of 12 months after hospital discharge and functional paramters like 6 minute walking test and pulmonary function were assessed.

The results stemming from the pulmonary function tests are in accordance with other works. However, the introduction is lacking in evidence and not offering a broad enough spectrum of why/how ARDS is affecting long-term pulmonary function. If this wasn't the case, why should COVID-19 associated ARDS treated with ECMO be different. THe authors are not clearly presenting a rationale for their study.

The results should be presented more clearly. For example, data on the NYHA status and its improvement over time could easily be plotted as bar graph with the different groups at the respective visitation time point. Thus, the data would easily convey the improvement visually, and not be buried in the text.

Why do the authors stress the vaccination status of their patients? The willingness/ability to be vaccinated should be commended, however, how does this affect pulmonary recovery? This part of the paper holds little to no merit. Furthermore, the patient sample is too small to draw any conclusions in regard to pulmonary recovery and previous vaccination status. As the authors state, vaccination prevents a complicated course, but does it really necessitate a non-favorable recovery?

Although the authors claim that they assessed health related quality of life, they failed to do so. THey only described some aspects of different modalities related to quality of life, but did so in an unstructured approach. Rather than merely describing the return to work rate, incidence of neurological complaints etc., the authors missed the opportunity to structurally investigate the different modalities by using an established  Score for health related quality of life (HADS, EQoL-5, WHODAS 2.0, SF36).

Overall the work gives a good insight into the longterm perspective and pulmonary recovery. However it is severely limited by the small sample size and methodological flaws.

minor:

some typograpical errors: for example p1 TLCO / DLCO (only the latter is used in the text); page 5 ll 175 lover should be lower. and a few more. nothing serious, but another spellcheck is recommended.

page 3: why Vancomycin prophylaxis? That appears to be excessive and not indicated.

table 2: why not list the stats for the total cohort? Where there any differences in the comorbidities between survivors and nonsurvivors?

page 4: as mentioned above, consider a graphical representation of the NYHA data

Author Response

Dear reviewer, thank you for your valuable review report. We adjusted the article according to your recommendations, which helped to improve article quality. Following part is aiming to answer or comment on your remarks.

In their work, Genzor et al. performed a small observational cohort study on 17 survivors of previous veno-venous ECMO treatment due to COVID-19 associated ARDS. The surviving patients were than observed over the period of 12 months after hospital discharge and functional paramters like 6 minute walking test and pulmonary function were assessed.

The results stemming from the pulmonary function tests are in accordance with other works. However, the introduction is lacking in evidence and not offering a broad enough spectrum of why/how ARDS is affecting long-term pulmonary function. If this wasn't the case, why should COVID-19 associated ARDS treated with ECMO be different. THe authors are not clearly presenting a rationale for their study.

Introduction was now adjusted – works about long-term sequelae of ARDS on lung functions were added

The results should be presented more clearly. For example, data on the NYHA status and its improvement over time could easily be plotted as bar graph with the different groups at the respective visitation time point. Thus, the data would easily convey the improvement visually, and not be buried in the text.

NYHA status bar plot was added

Why do the authors stress the vaccination status of their patients? The willingness/ability to be vaccinated should be commended, however, how does this affect pulmonary recovery? This part of the paper holds little to no merit. Furthermore, the patient sample is too small to draw any conclusions in regard to pulmonary recovery and previous vaccination status. As the authors state, vaccination prevents a complicated course, but does it really necessitate a non-favorable recovery?

We do not think, that vaccination after the covid-19 affects the recovery. The data about the vaccination were added in aim to share data for further possible metanalyses.  

Although the authors claim that they assessed health related quality of life, they failed to do so. THey only described some aspects of different modalities related to quality of life, but did so in an unstructured approach. Rather than merely describing the return to work rate, incidence of neurological complaints etc., the authors missed the opportunity to structurally investigate the different modalities by using an established  Score for health related quality of life (HADS, EQoL-5, WHODAS 2.0, SF36).

We agree, the comments on this part was added in the discussion in the section Study limitations.

Overall the work gives a good insight into the longterm perspective and pulmonary recovery. However it is severely limited by the small sample size and methodological flaws.

We agree, some remarks were added in the section Study limitations

minor:

some typograpical errors: for example p1 TLCO / DLCO (only the latter is used in the text); page 5 ll 175 lover should be lower. and a few more. nothing serious, but another spellcheck is recommended.

Adjustement performed

page 3: why Vancomycin prophylaxis? That appears to be excessive and not indicated.

Comments were added in the discussion – low dose vancomycin prophylaxis (125 mg p.o. per day) was administered according to the local (Czech) microbiological society recommendation. Refference about the justification of this therapy was added.

table 2: why not list the stats for the total cohort? Where there any differences in the comorbidities between survivors and nonsurvivors?

Unfortunately, we do not have full data about all of the individuals, frequently were patients reffered from peripheral hospitals and the data is simply missing. This was added in the study limitations section.

page 4: as mentioned above, consider a graphical representation of the NYHA data

Agree, added